# Transcriptomic Analysis Reveals Insect Hormone Biosynthesis Pathway Involved in Desynchronized Development Phenomenon in Hybridized Sibling Species of Tea Geometrids (*Ectropis grisescens* and *Ectropis obliqua*)

**DOI:** 10.3390/insects10110381

**Published:** 2019-11-01

**Authors:** Zhibo Wang, Jiahe Bai, Yongjian Liu, Hong Li, Shuai Zhan, Qiang Xiao

**Affiliations:** 1Key Laboratory of Tea Quality and Safety Control, Tea Research Institute, Ministry of Agriculture, Chinese Academy of Agricultural Sciences, Hangzhou 310008, China; wangzhibo9527@163.com (Z.W.); baijhtricaas@163.com (J.B.); hongli_tea@163.com (H.L.); 2Key Laboratory of Insect Developmental and Evolutionary Biology, Institute of Plant Physiology and Ecology, Shanghai Institutes for Biological Sciences, Chinese Academy of Sciences, Shanghai 200032, China; liuyongjian@sippe.ac.cn (Y.L.); szhan@sibs.ac.cn (S.Z.)

**Keywords:** sibling tea geometrid, hybridization generation, desynchronized development, transcriptome analysis, Juvenile hormone, moulting hormone

## Abstract

*Ectropis grisescens* and *Ectropis obliqua* are sibling species of tea-chewing pests. An investigation of the distribution of tea geometrids was implemented for enhancing controlling efficiency. *E. grisescens* is distributed across a wider range of tea-producing areas than *Ectropis obliqua* in China with sympatric distribution found in some areas. In order to explore reproductive isolation mechanisms in co-occurrence areas, hybridization experiments were carried out. Results showed they can mate but produce infertile hybrids. During experiments, the desynchronized development phenomenon was found in the hybridized generation of sibling tea geometrids. Furthermore, transcriptome analysis of those individuals of fast-growing and slow-growing morphs revealed that the insect hormone biosynthesis pathway was enriched in two unsynchronized development groups of hybrid offspring. More importantly, some genes regulating the synthesis of moulting hormone showed significantly up-regulated expression in fast-growing groups. Above all, metabolism of the juvenile hormone and synthesis of the ecdysone pathway were found to be crucially involved in the desynchronized development phenomenon. This research finding contributes to a better understanding of the mechanisms of insect development and reproductive isolation of two sibling species.

## 1. Introduction

Tea is an important non-alcoholic beverage crop [1]. Hence, improving the yield and quality of tea would provide meaningful economic gain for many countries, particularly China, where tea is cultivated in [2]. China is the largest tea-growing country, in which the tea plantation area reached approximately 3.1 million hectares in 2017 and currently includes 20 provinces and regions [3]. In China, two sibling tea geometrid species, *Ectropis grisescens* (Warren) (Lepidoptera: Geometridae, Ennominae) and *Ectropis obliqua* (Prout), are the most serious pests chewing leaves of tea [4,5]. The larvae feed on tea leaves and tender buds, resulting in a severe reduction in tea production and quality in the current and following years [4,6].

Though *Ectropis grisescens* and *Ectropis obliqua* were named by Warren and Prout several decades ago [7,8], the two sibling tea geometrids were always mistaken as one species because of their extremely similar morphology [9]. Such mistakes also caused a lot of problems in terms of prevention and control for tea-plant management. Afterwards *E. obliqua* nucleopolyhedrovirus (*Eo*NPV), an effective natural biological agent, has been successfully developed as a bio-insecticide in China for the control of the tea geometrids [4,10]. Following that, Xi et al. found that some tea geometrid populations revealed diverse sensitivity to the virus but they did not know the reason at that time [10]. Jiang et al. successfully differentiated the tea geometrid species in China as two independent species, i.e., *E. grisescens* and *E. obliqua*, based on evidence of morphology and molecular biology [5]. Specifically, the 3.7% genetic distance between *E. obliqua* and *E. grisescens*, based on cytochrome oxidase I (COI) sequences, was found to be a practical distinguishing method that can provide a basis for further research on sibling species of tea geometrids. At the same time, they also found the reason why virus sensitivity showed that, despite differences between the two species, *E. obliqua* is more sensitive to *Eo*NPV than *E. grisescens* [5,11]. Further research focused on sibling species of tea geometrid after they became easily distinguished. Another bio-control method of sex pheromone trapping the sibling species of tea geometrids was restarted with the different components of sex pheromone reported between them [12]. The main sex pheromone components of *E. grisescens* were Z3, Z6, Z9-18:H and Z3, epo6, Z9-18:H, correspondingly Z3, Z6, Z9-18:H, Z3, epo6, Z9-18:H and Z3, epo6, Z9-19:H were sex pheromone components of *E. obliqua* [13]. As sibling species, tea geometrids sharing the same effective components of sex pheromone mean that reproductive interference may exist in sympatry. In fact, Zhang et al. verified the hypothesis of asymmetrical reproductive interference existing between two sibling geometrids in the laboratory [14]. Males of both *E. grisescens* and *E. obliqua* could interfere through the intraspecific mating of the other species, but the impact of the mating interference differed. Such asymmetric mating preference and competitive interaction might be a driving force of reproductive isolation, speciation and evolution [15,16].

In order to accurately control to the two-sibling tea geometrids, their distribution was investigated in China and sympatric distribution of two species was found in some areas. Geographical isolation can be a pathway of reproductive isolation. For the two-sibling tea geometrids, it was not clear whether reproductive isolation existed in the condition of incomplete geographical isolation. Furthermore, hybridization experiments were implemented in order to verify this. During the study, transcriptomic analysis was applied for those asynchronously growing hybrid individuals in order to explore the regulating mechanism of desynchronized development phenomenon. More importantly, we think the filial generation is appropriate material for researching on the mechanism of insect development and reproductive isolation.

## 2. Materials and Methods

### 2.1. Sample Collection

Larvae of tea geometrids were captured from 11 Chinese provinces that represent the main tea-producing areas in China (Appendix A). At least 10 samples were collected from each location.

### 2.2. Sample Identification

The specie identification of tea geometrids was confirmed by the sequence analysis of the COI gene. A total of 757 specimens were sequenced for this study. Genomic DNA was extracted from each individual, either a larva or an adult, using a TIANamp Genomic DNA kit (Tiangen, Beijing, China). Prepared DNA was dissolved in 100 µL deionized and sterilized water and stored at −80 °C for later use. Polymerase chain reaction (PCR) amplification of the COI gene was performed using the primers COI-F: 5′-ATTCAACCAATCATAAAGATATTGG-3′ and COI-R: 5′-CTCCWCCAGCAGGATCAAAA-3′ and a previously described procedure [5]. PCR products were determined using the Sanger sequencing method. Contigs of each sample were independently assembled using SeqMan 5.01 (Madison, WI, USA). Multiple alignments of assembled COI sequences were compared with reference sequences downloaded from the National Center for Biotechnology Information using Clustal X 1.83 (EMBL, Heidelberg, Germany). Phylogenetic analyses were conducted to identify species using a maximum likelihood (ML) approach implemented in MEGA 5.0 with 1000 bootstrap replicates.

### 2.3. Hybridization Experiment

We randomly selected single populations of *E. obliqua* (Yuhang population) and *E. grisescens* (Xinchang population) for further crossing experiments. The two populations were separately reared in different areas of phytotron (temperature 24–26 °C, humidity 50%–70%, photoperiod L14:D10). After pupation, male and female pupae were separated based on morphological characters. One day after emergence, adult moths were randomly collected for crossing experiments. The crossing experiments included four combinations which were two interspecific combinations (*E. grisescens*♀ × *E. obliqua*♂; *E. grisescens*♂ × *E. obliqua*♀) and two intraspecific combinations (*E. grisescens*♀ × *E. grisescens*♂; *E. obliqua*♂ × *E. obliqua*♀). The crossing experiments were implemented at 09:00 am. A pair of moths were put into a colourless glass cage. Female moths were allowed to lay eggs for 5 days and then removed. Eight replicates were used for each treatment. Larvae of the F1 generation were reared to pupae and larval and pupae developmental duration was measured.

### 2.4. Sample Preparation and Collection for Transcriptomic Analysis

Filial generation of *E. grisescens*♀× *E. obliqua*♂ was used in this transcriptomic analysis. Each group included three individuals as biological duplication. The larvae of the F1 generation which hatched on the same day were selected and fed with fresh leaves under the same conditions. The 2nd instar larvae (Code name: C), 5th instar larvae (Code name: L), early pupa at 13th day (Code name: E) and late pupa at 30th day (Code name: F) were sampled for transcriptomic analysis. All tissues were immediately snap-frozen in liquid nitrogen, and stored at −80 °C until extraction.

### 2.5. Total RNA Extraction, cDNA Library Construction and Illumina RNA-Seq

Total RNA was extracted using TRIzol regent (Life Technologies, Carlsbad, CA, USA) according to the manufacturer’s protocol. One percent agarose gel electrophoresis was used to confirm the integrity of the RNA. The quality of the RNA was further assessed using the RNA Nano 6000 Assay Kit of the Bioanalyzer 2100 system (Agilent Technologies, Santa Clara, CA, USA).

A total amount of 1.5 μg RNA per sample was used as input material for the RNA sample preparation for transcriptome sequencing. The sequencing libraries were generated using a NEBNext^®^ Ultra™ RNA Library Prep Kit for Illumina^®^ (Illumina, San Diego, CA, USA). Poly-T oligo-attached magnetic beads were used to purify the mRNA from the total RNA. Afterwards, the first-strand cDNA and second-strand cDNA were synthesized, respectively.

According to the manufacturer’s instructions, the cBot Cluster Generation System using TruSeq PE Cluster Kit v3-cBot-HS (Illumina) was used to perform clustering of the index-coded samples. Subsequently, the library preparations were sequenced on an Illumina HiSeq platform and paired-end reads were generated.

### 2.6. Transcriptome Assembly and Functional Annotation

The clean reads were filtered from the raw reads by quality analysis using Fast QC which removed reads containing adaptor contamination, low-quality bases and undetermined bases using trimmomatic. The total clean reads from four libraries in each group were pooled and subjected to de novo assembly using Trinity. After that, Benchmarking Universal Single-Copy Orthologs (BUSCO) and perl scripts in Trinity were performed to assess the assembly quality.

The majority of functional annotation was followed the pipeline recommendation of Trinotate software (Broad Institute, Cambridge, MA, USA). Initially Open Reading Frame (ORF) prediction was done by Transdecoder, that both transcripts and predicted peptides were aligned against the Swissprot and the Pfan-A databases through Blast and HMMER. Later the result data, together with transcripts and gene-trans map file were imported to the sqlite database which aims to find the association with Gene Ontology (GO), Kyoto Encyclopedia of Genes and Genomes (KEGG), and eggnog databases. Finally, Trinotate was used to integrate the annotation results.

### 2.7. Differentially Expressed Genes (DEGs) Analyzes and Gene Ontology (GO), KEGG Enrichment of DEGs

Bowtie2 aligner was used to map the clean reads to the *de novo* generated transcriptome. Quantitative analysis of gene expression was finished by RNA-Seq by Expectation-Maximization (RSEM). Differential expression analysis of two conditions/groups was performed using the DESeq2 R package 1.24.0 (Robert Gentleman and Ross Ihaka, University of Auckland, New Zealand). Genes with a statistical significance (*p*-value < 0.05) and absolute value of log_2_ (fold change) > 1 found by DESeq2 were assigned as differentially expressed. All DEGs between stages were further subjected to GO enrichment analysis and KEGG pathway enrichment analysis to verify biological significance through cluster profiler package in R.

### 2.8. Analyzes of Genes Related to Developmental Patterns

The protein sequences of *Bombyx mori* (silkworm) in juvenile hormone (JH) and moulting hormone (ecdysone) pathway and transcripts were aligned with each other through tBlastn and Blastp, with the best match transcripts regarded as canditate genes of the JH and ecdysone pathways. Significant differentially expressed genes were calculated through the pipeline of DESeq2, and fragments per kilobase of transcript per million mapped reads (FPKM) were visualized by using a pheatmap package in R.

### 2.9. Statistical Analysis

Normal data distribution was tested for pupal duration of offspring in the hybridization experiment using the Kolmogorov–Smirnov test in SPSS.

## 3. Results

### 3.1. Distribution of the Two Closely Related Tea Geometrid Species

We conducted sampling and identification throughout China. All 757 specimens of 19 populations collected from 11 provinces were identified based on genetic divergence in the mtDNA COI region (~650 bp) and phylogenetic tree (Figure 1), and their geographical distribution was defined. A total of 29 sequences for *E. grisescens* and *E. obliqua* were deposited in GenBank (Appendix A). Based on our results, *E. grisescens* is distributed in 11 locations in a wide range of tea-planting areas in China; *E. obliqua* is distributed in only 3 locations (Huzhou, Yuhang, and Wuxi) on the borders of Zhejiang, Jiangsu and Anhui provinces; and the two species coexist in 5 locations including Xuancheng of Anhui province, Liyang of Jiangsu province and Linan, Fuyang, Tea Research Institute (TRI) of Zhejiang province (Figure 2).

### 3.2. Development Situation of Filial Generation

The study of geographical distribution showed the two species were sympatric distributed in some areas, which meaning hybridization may exist in field. Subsequently, we did crossing experiments to verify the reproductive isolation between the two species. The results showed that the interspecific combinations *E. grisescens*♀ × *E. obliqua*♂ and *E. grisescens*♂ × *E. obliqua*♀ both can mate but produce infertile hybrids. During development of filial generation, we found an interesting phenomenon that larvae of *E. grisescens*♀ × *E. obliqua*♂ did not keep synchronous growth. For comparison, the pupal duration of intraspecific combinations was distributed within a time period (approximately 10 days) showing a normal distribution trend (*p* > 0.05), while the pupal duration of *E. grisescens*♀ × *E. obliqua*♂ (24 days) is longer than other 3 combinations and did not have a normal distribution trend (*p* = 0.01) (Figure 3). To give a typical example, there were 2nd instar larvae, mature larvae, pupa and even adults present at the same time for those filial generations of *E. grisescens*♀ × *E. obliqua*♂.

### 3.3. Summary of RNA-Seq

12 RNA samples from four groups (2nd instar larvae, 5th instar larvae, early pupa at 13th day and late pupa at 30th day) were sequenced with three replicates included in each group. A total of 2,723,883,472 raw reads were generated from four libraries. After filtering the low quality and adaptor sequences, 574,716,928; 607,827,264; 749,191,768; and 735,900,560 clean reads were obtained in each group, respectively. Of the clean reads, the Q20 percentage (proportion of sequences with a sequencing error rate less than 1%) for all samples exceeded 97%.

All samples of larvae were assembled together into 268,389 transcripts with an average length of 1026 bp. While all samples of pupae were assembled together into 316,568 transcripts with an average length of 1007 bp (Figure 4).

Using the Trinity de novo assembly programme, larvae and pupa group of all short-read sequences were respectively assembled into 168,099 and 199,403 unigenes (Table 1). All assembled unigenes were aligned against different protein databases included GO, KEGG, Pfam, eggNOG, and Swiss-Prot.

### 3.4. Analysis of DEGs

Differences in gene expression during development of the filial generation were examined, and two different methods were adopted. First, pairwise comparison between groups of C and L was undertaken, in order to search the DEGs in those samples with the same development times but different developmental instar in the larval stage. Another comparison between E and F was also carried out, in order to search the DEGs in those samples with same development starting point and stage but different developmental duration. A total of 5337 DEGs was found between the C and L libraries, including 2031 up-regulated and 3306 down-regulated genes (Figure 5A). In comparisons of pupae, a total of 8041 DEGs were found between E and F, including 3337 up-regulated and 4704 down-regulated genes (Figure 5B).

### 3.5. Functional Classification of DEGs during Development of Filial Generation

In order to identify the major transcriptome changes for desynchronized development of filial generation, we performed studies of enrichment of DEGs in GO and KEGG functional categories in pairwise comparisons of C-vs-L and E-vs-F.

In the C-vs-L, only 1853 DEGs can be annotated by the GO database based on sequence homology, of which 781 genes can be enriched in 86 GO terms (Appendix A). Among these terms “motile cilium”, “serine-type endopeptidase activity” and “odorant binding” were the most abundant. While in another comparison of E-vs-F, a total of 2626 DEGs were annotated by the GO database, with 1814 genes enriched in 547 GO terms that involved the most significant subsets named “structural constituent of cuticle”, “motile cilium” and “cilium” (Appendix A). Enriched GO terms in two groups are shown in Figure 6A,B. After taking the intersection of these comparisons, there were 54 enriched subsets that could be found in both groups, which were divided into 3 categories named MF (molecular function), BP (biological process) and CC (cellular component). In the molecular function category, the GO term “serine-type endopeptidase activity” is the most significantly enriched process which includes 84 genes in C-vs-L group and 79 genes in E-vs-F group; following that a number of genes were enriched in “structural constituent of cuticle”, “odorant binding” and other subsets. In the biological process category, many genes in two groups were enriched in the “ion transport” term. In the cellular component category, the majority of genes were illustrated to be related to “motile cilium”, “axoneme”, “organelle membrane”, “cilium” and “lysosome” GO terms. All enriched subsets of both groups are listed in Appendix A.

KEGG pathway enrichment analysis was used to verify biological significance. In C-vs-L and E-vs-F comparisons, 974 and 1411 DEGs were annotated using the KEGG database. In two comparisons, both have 53 genes significantly enriched in 6 and 7 KEGG pathways, respectively (Figure 5). In the C-vs-L comparison, the most enriched pathway was “Non-alcoholic fatty liver disease (NAFLD)”, followed by the longevity-regulating pathway—worm, pentose phosphate pathway, Toll and Imd signaling pathways, insect hormone biosynthesis and tyrosine metabolism (Figure 7A). The pentose phosphate pathway, longevity-regulating pathway—worm, lysosome, amoebiasis, retinol metabolism, fat digestion and absorption and insect hormone biosynthesis, in order, were the most enriched pathways in E-vs-F comparison (Figure 7B). Among those pathways, longevity-regulating pathway—worm, insect hormone biosynthesis and pentose phosphate pathway were enriched in both C-vs-L and E-vs-F.

### 3.6. Identification of Developmental-Related DEGs

Juvenile hormone (JH) and moulting hormone (ecdysone) are most important insect hormones in the regulation of development. In order to identify development-related genes in desynchronized development of the filial generation, BLAST against homologous protein sequences of silkworm was performed, and we investigated 23 genes with homology to known developmental-related genes in JH and ecdysone pathways from the KEGG database. In the C-vs-L comparison, we found 6 significant differences in genes HMG_CoA_reductase (HMGR), diphosphomevalonate_decarboxylase (MevPPD), farnesyl_diphosphate_phosphatase (FPPP), farnesol_dehydrogenase (FDP), juvenile_hormone_diol_kinase (JHDK) and ecdysteroid_22-hydroxylase (Dib) in JH and ecdysone pathways (Figure 8A, Appendix A). Among them, HMGR, FDP and Dib genes had up-regulated expression, while MevPPD, FPPD and JHDK had down-regulated expression in L group. In addition, we also found 6 significant differences among genes HMG_CoA_synthase (HMGS), phosphomevalonate_kinase (MevPK), FDP, JHDK, Dib and 26-hydroxylase (CYP18A1) in the E-vs-F comparison, which of HMGS, MevPK and CYP18A1 had up-regulated expression (Figure 8B, Appendix A).

## 4. Discussion

Speciation is a process of evolution which is attributed to many theories and hypotheses by different scientists [17]. The dominant factor of natural selection and the prerequisite of isolation are the generally accepted point of views [17,18]. Thus, geographical distribution is an important aspect for deducing sibling species formation and differentiation. The results revealed an arc-shape coexisting areas between distribution of the two species, which means geographical isolation has not been completed yet. Although the two species tea looper seemed to occupy their own territories, competition must exist in those overlapping areas. Competition may shape evolutionary processes between closely related species and involve multiple mechanisms, such as differential biological characteristics [19,20], endosymbiotic bacteria [21], and reproductive interference [22].

Commonly, reproductive isolation is a vital symbol of speciation. The conventional definition is that different species cannot mate and breed offspring [23]. In our study, the result of hybridization experiments illustrated that sibling species tea looper can mate but breed the infertile hybrids. This is an uncommon phenomenon which is rare in another sibling species, such as *Helicoverpa armigera* and *Helicoverpa assulta*, and *Heliothis virescens* and *Heliothis subflexa* [24,25]. The phenomenon was named postzygotic reproductive isolation [26]. It suggested that the sibling species tea looper may be at the transitional stage of speciation.

For further study, we concerned ourselves with the desynchronized development phenomenon in the hybridized F1 generation of sibling species of tea geometrids and ascertained the mechanism of the desynchronized development to better understand how reproductive isolation occurs. We sequenced the genes and performed transcriptome analysis on individuals with differential development (slow and fast). Furthermore, we screened the relevant key genes from two compared groups. GO functional categories showed DEGs were enriched in both C-vs-L and E-vs-F. We found that the majority of enriched subsets in both groups were in molecular function and biological process categories, of which many were related to construction of cell structure, enzyme activity and biomass transport. These results suggested that there were several varieties in gene functions between different inchworm development stages, which might contribute to the desynchronized development phenomenon. Meanwhile KEGG pathway enrichment analysis showed the longevity-regulating pathway—worm, pentose phosphate pathway and insect hormone biosynthesis pathway were enriched in both C-vs-L and E-vs-F. Longevity-regulating pathway involves lifespan in both worms and mammals. It is coordinated by intersecting pathways, often converging on metabolic networks [27]. A key player in lifespan regulation is the mitochondrion which is the site of ATP synthesis [28]. The development of insects needs ATP from respiration. Therefore, the longevity-regulating pathway is related to energy supply during insect development. The pentose phosphate pathway is active in a wide variety of cell types, particularly those of which that have a high rate of nucleotide synthesis or which utilize NADPH in large amounts [29]. A mass of nucleotide synthesis occurred during periods when tissues rapidly divide. NADPH is needed in the biosynthesis of steroid hormones such as molting hormone with the function of regulating growth of insects [30]. The insect hormone biosynthesis pathway is vitally important for our study. Among those hormones, JH and ecdysone are most important in regulating development [31,32]. We investigated 23 developmental-related genes in juvenile hormone and ecdysone pathways and found 6 significant differences in genes in C-vs-L and E-vs-F, respectively. In fact, synthesis of JH and ecdysone needed the same precursor substance—farnesyl pyrophosphate (FPP) which was synthesized in the mevalonate (MVA) pathway [33]. We found some genes of the MVA pathway such as HMGR, HMGS and MevK show up-regulated expression in the fast-growing groups. This suggested that more FPP can be utilized in ecdysone synthesis. In Juvenile hormone pathway, the genes FDP and JHDK show significant differences both in C-vs-L and E-vs-F groups (Figure 8). In particular, JHDK show up-regulated expression in two slow-growing groups. JHDK is an enzyme crucial for secondary metabolism of JH. It can catalyze the conversion of JH diol to JH diol phosphate [34]. This suggested that JHDK may play a role in the temporal suppression of JH titer and regulate development of slow-growing groups indirectly.

In the ecdysone pathway, we found Dib and CYP18A1 significantly up-regulated expression for the fast-growing groups in C-vs-L and E-vs-F, respectively. Although Dib also showed significant differences in E-vs-F, the down-regulated expression may not explain for fast-growing groups. Previous studies showed the three P450 genes, phm (CYP306a1), Dib (CYP302a1) and Sad (CYP315a1), directly mediate the last three hydroxylation reactions in the synthesis of Ecdysone [35]. Phm encode the microsomal 25-hydroxylase which can catalyze 2,22,25-trideoxyecdysone (3b,5b-ketodiol) conversion to 2,22-dideoxyecdysone (3b,5b-ketotriol). Then, 2,22-dideoxyecdysone was catalyzed into 2-deoxyecdysone by the 22-hydroxylase (encoded by Dib). Sad is responsible for the conversion of the 2-deoxyecdysone to ecdysone [36]. All of them were the key gene in the ecdysone pathway. In brief, the expression of Dib can regulate the titers of ecdysone, which influence the development of insects.

In this study, we searched for the cause of the desynchronized development phenomenon in the hybridized generation of sibling species of tea geometrid by two different comparisons. As a result, some developmental genes were found in each comparison. Although those significantly different genes were not completely consistent in the two comparisons, which may be attributed to the different developmental stage of samples, JH and ecdysone were both involved in the desynchronized development phenomenon. The mechanism can be explained the following way: the development of the insect was synergistically regulated by JH and ecdysone. In fast-growing groups, the more activity in the MAV pathway resulted in more synthesis of FPP which provided the basis for the synthesis of ecdysone. In a normal synthesis level of juvenile hormone, metabolism of JH and synthesis of ecdysone can explain the desynchronized development phenomenon of the hybridized generation of sibling species of tea geometrid. To synthesize the results, we offer a hypothesis that the sibling tea looper species are at an early stage of speciation because of the relatively independent distribution and incomplete reproductive isolation.

## 5. Conclusions

The distribution investigation of sibling tea geometrid discovered instances of sympatry. Thereby, the hybridization experiments between *E. obliqua* and *E. grisescens* were implemented and the desynchronized development phenomenon was found in the hybridized F1 generation of sibling tea geometrid species. Transcriptomic analysis illustrated that the insect hormone biosynthesis pathway was enriched in two unsynchronized development groups of hybrid offspring. Finally, we found key genes JHDK and Dib involved in the metabolism of juvenile hormone and synthesis of the ecdysone pathway regulating the development of those hybridized individuals showing desynchronized development by transcriptome analysis. Those individuals showing desynchronized development can be treated as the material for research into the mechanism of insect development and reproductive isolation. At the same time, it also may offer some clues toward the development of new technology for pest management.

## Figures and Tables

**Figure 1 insects-10-00381-f001:**
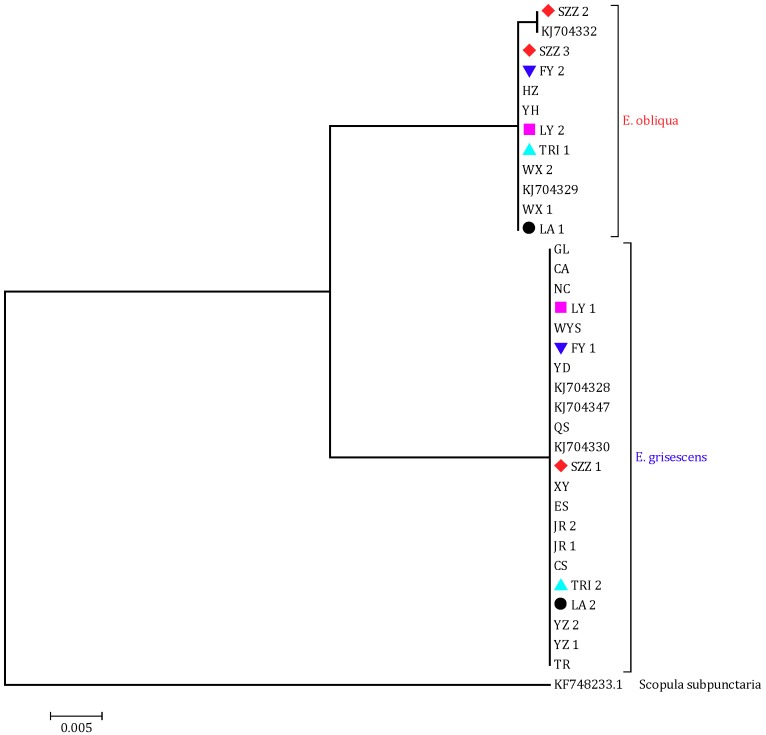
The maximum likelihood (ML) tree of tea geometrids and the outgroup species based on cytochrome oxidase I (COI) sequences. Marks in the figure mean the co-existant areas and the same mark in one place.

**Figure 2 insects-10-00381-f002:**
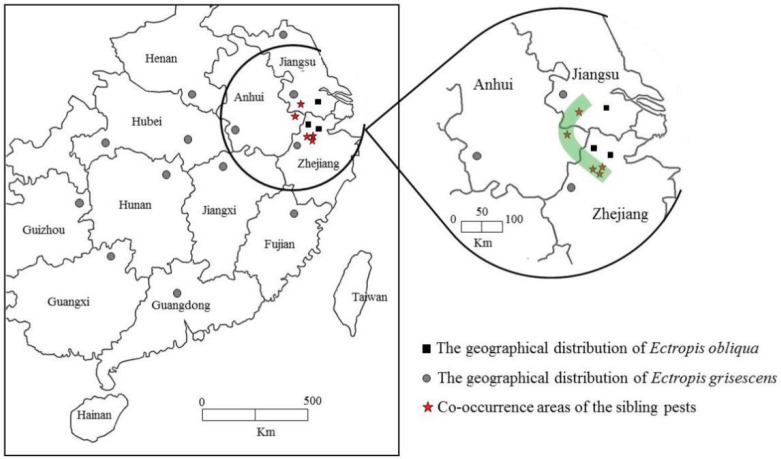
Sampling locations in China. Grey dots are the distribution areas of *E. grisescens*, and black squares are the distribution areas of *E. obliqua*. The co-occurrence of *E. grisescens* and *E. obliqua* is indicated by red stars and green shading.

**Figure 3 insects-10-00381-f003:**
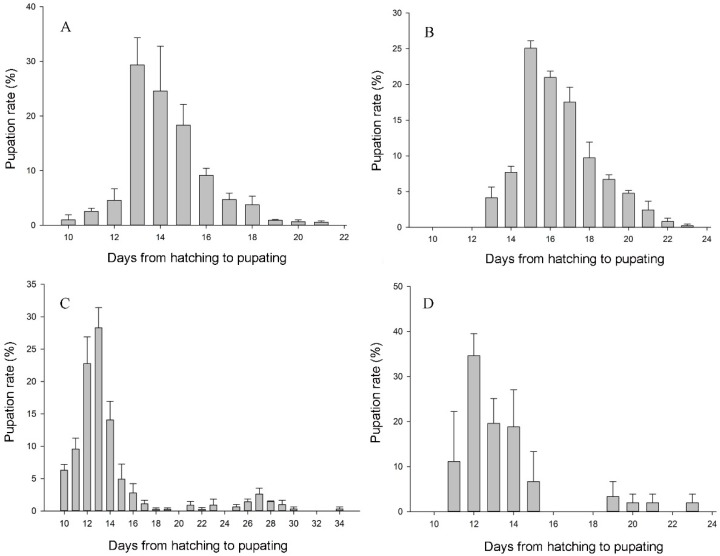
Temporal distribution of pupation in F1 generation of hybrids of *E. grisescens* and *E. obliqua*. (**A**) *E. grisescens*♀ × *E. grisescens*♂; (**B**) *E. obliqua*♀ × *E. obliqua*♂; (**C**) *E. grisescens*♀ × *E. obliqua*♂; (**D**) *E. grisescens*♂ × *E. obliqua*♀. Error bars represent standard error of the means.

**Figure 4 insects-10-00381-f004:**
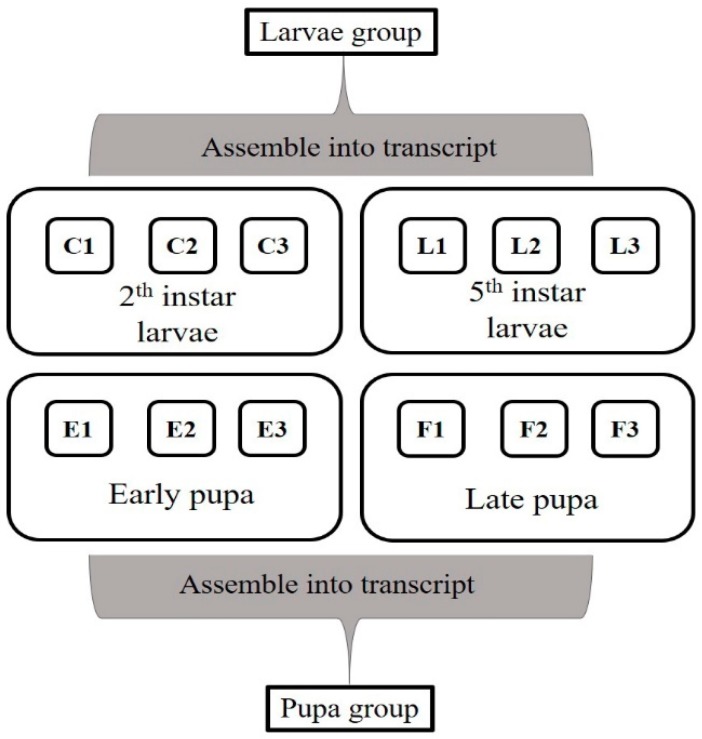
Transcripts of all samples were assembled to two groups. Larvae group (C and L) were assembled from all larvae samples; pupa group (E and F) were assembled from all pupa samples. C: 2nd instar larvae; L: 5th instar larvae; E: early pupa; F: late pupa. All samples hatched on same day. Samples of C, L and E were sampled on the 13th days; samples of F were sampled on the 30th days.

**Figure 5 insects-10-00381-f005:**
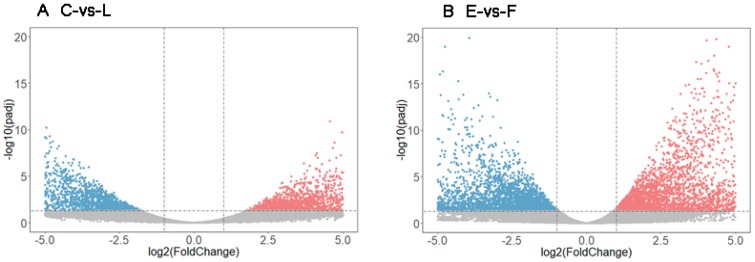
Volcano plot of differentially expressed genes (DEGs) in C-vs-L group (**A**) and E-vs-F group (**B**). The horizontal ordinate represents the fold change (log2 FC) of gene expression in the different experimental groups, and the vertical ordinate represents statistical significance (−log10 P adj) of the change of gene expression. Each point in the plot represents each gene, and the red and blue points represent the significant up and down-regulated genes.

**Figure 6 insects-10-00381-f006:**
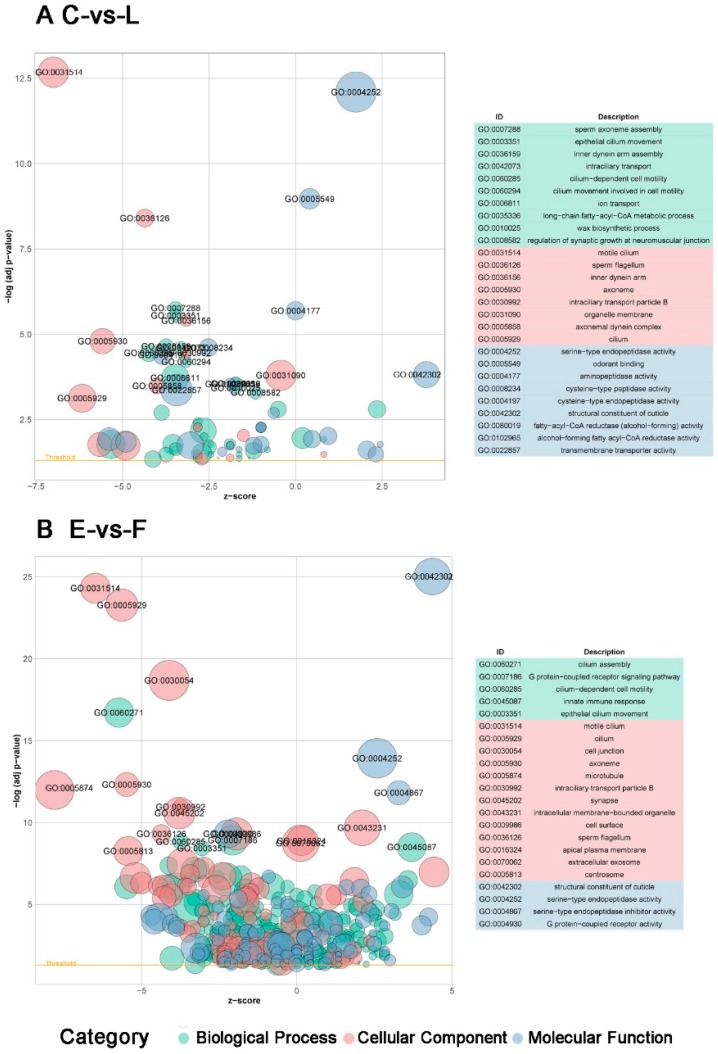
Gene Ontology (GO) enrichment terms of DEGs in C-vs-L group (**A**) and E-vs-F group (**B**). GO terms grouped into three main categories: biological process, cellular component and molecular function. The X-axis means Z-score, that illustrates whether a GO term of experiment group is likely increased (positive values) or decreased (negative values) compared with the control group. The size of circle shows the number of genes involved in each GO term and colours represent variant categories. The yellow line demarcates the significance level of adjusted *p*-value (0.05).

**Figure 7 insects-10-00381-f007:**
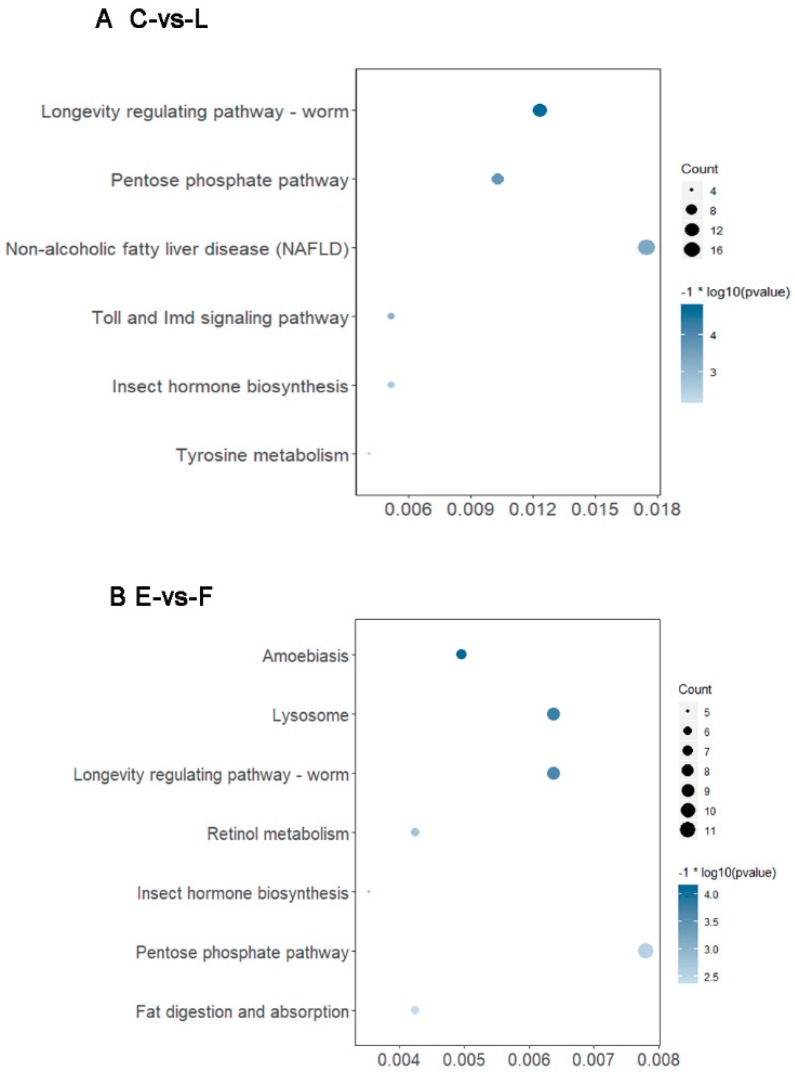
Analysis of KEGG pathway enrichment in C-vs-L group (**A**) and E-vs-F group (**B**). The horizontal ordinate represents gene ratio in each enriched pathway. The larger dot indicates the greater number of genes, and the deeper blue color indicates the larger *p*-value.

**Figure 8 insects-10-00381-f008:**
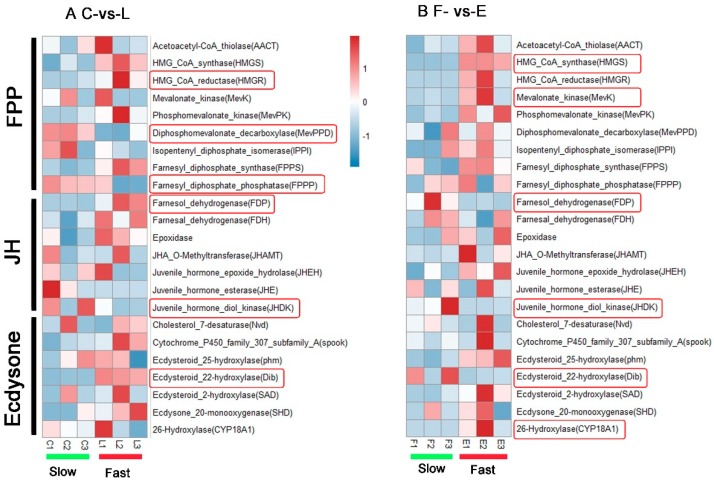
Heatmap of normalized gene expression (fragments per kilobase of transcript per million mapped reads (FPKM) value) in juvenile hormone (JH) and moulting hormone (ecdysone) pathways in C-vs-L group (**A**) and E-vs-F group (**B**). Red indicates high expression and blue signifies low expression. The significant differentially expressed genes evaluated by DESeq2 were circled out by rounded rectangles.

**Table 1 insects-10-00381-t001:** Summary of transcriptome in each sample after quality control.

Sample	Raw Reads	Clean Reads	Valid (%)	Q20 (%)	Q30 (%)	GC (%)
C1	52,103,142	50,962,816	97.81	97.88	94.11	51.64
C2	50,825,206	49,705,732	97.80	97.56	93.46	52.31
C3	44,598,296	43,010,684	96.44	97.48	93.45	55.64
L1	46,276,016	45,479,152	98.28	97.91	94.02	51.80
L2	54,087,972	52,918,552	97.84	97.21	92.42	51.85
L3	55,174,098	53,559,112	97.07	97.60	93.32	54.47
E1	59,386,648	58,378,284	98.30	97.21	92.49	46.61
E2	56,732,666	55,961,382	98.64	97.24	92.57	47.98
E3	74,080,310	72,958,276	98.49	97.35	92.77	47.70
F1	63,019,000	61,842,676	98.13	97.24	92.57	47.51
F2	63,144,758	62,126,664	98.39	97.15	92.34	46.71
F3	61,542,756	60,005,800	97.50	97.34	92.81	46.32

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
