# Peer review of "Transcriptomic Analysis Reveals Insect Hormone Biosynthesis Pathway Involved in Desynchronized Development Phenomenon in Hybridized Sibling Species of Tea Geometrids (Ectropis grisescens and Ectropis obliqua)"

_insects, 2019, doi:10.3390/insects10110381_

Round 1

Reviewer 1 Report

Here is my review report on “Transcriptomics reveals moulting hormone involved in desynchronized development phenomenon in hybridized sibling species of tea geometrids (Ectropis grisescens and Ectropis obliqua )” (insects-601444), submitted for publication consideration to Insects. For the above reasons I suggest a major revision of this manuscript to support the conclusions (see major comments) or at least extensively revised the conclusions and discussed further works that will be necessary to support the proposed hypothesis.

Major: In my opinion the conclusions are not supported by the data. In absence of functional validation (first qPCR validation of the transctiptiomic data) and at least a functional validation of some key genes’ confirmation and/ or hormone quantitation the conclusions are not validated since a lot of things could happened from a simple difference of gene expression and a physiological response.

Minor: The different scale for x axis of Figure 2 could be disturbing for the reader (as for the reviewer).

Author Response

Dear reviewer:

      I am very appreciating your review, and answer your questions as fellow:

Reviewer1: In my opinion the conclusions are not supported by the data. In absence of functional validation (first qPCR validation of the transctiptiomic data) and at least a functional validation of some key genes’ confirmation and/ or hormone quantitation the conclusions are not validated since a lot of things could happened from a simple difference of gene expression and a physiological response.

Answer: Thanks for your advice which offering a help for our future research. In this article, we showed the distribution of the sibling species, desynchronized development phenomenon in hybridized and screened the related genes which were responsibility for the phenomenon. Recently, we have begun to doing some work which of expression difference for those hormone and related genes in developing slow and fast simples. The result will help us to find more accurate reason for the phenomenon including the location of developmental stage and key gene.

Reviewer1: The different scale for x axis of Figure 2 could be disturbing for the reader (as for the reviewer).

Answer: we amend the scale of x axis in Figure 2. Now, the scale of the four results were same in Figure 2. In addition, we also changed the sequence of the four results of Figure 2 in order to have a better understanding for readers.

Reviewer 2 Report

I received and reviewed the article titled “Transcriptomics reveals molting hormone involved in desynchronized development phenomenon in hybridized sibling species of tea geometrids (Ectropis grisescens and Ectropis obliqua)” communicated for publication in the journal “Insects”. The article is quite informative and shares some essential information. But, there persists a many issues and queries which in my opinion must be sorted out to make article more clear and justfied.

Article title doesn’t read well. Try to improve if possible like changing “Transcriptomics” I to “Transcriptomic analysis”. Lines 52, not clear what COI stands for? Make sure to use full forms of any abbreviations before denoting this”. In my opinion, the final lines of introduction (line 76-77) belong to conclusion section. Modify the line 85, Cytochrome Oxidase I (COI) needs to be mentioned before rather than here. I presume this article needs more information on statistical tests done and threshold used. Like, information of line 109-110, Normal data distribution was tested using Kolmogorov Smirnov test in SPSS should be moved to formal statistical analysis section. The line 112 states that filial generation of E. grisescens ♀ × obliqua ♂ was used in this transcriptomic analysis. Any logical reason to select and use this cross and filial generation rather than E. grisescens ♀ × E. grisescens ♂? As mentioned in line 148, authors chosen absolute value of log 2 (fold change) > 1 as threshold for DEG’s. Any specific reasons? As per general understanding the people use >2 fold change as threshold for this kind of analysis. Again, figure 2 doesn’t explain enough the king of error bars used in graphing. I hope to include more information related graphing in figure legends. Also in my opinion, sequence of figure needs to be changed to make it more presentable. Like, C & D should be A & B while A & B should be C & D. Figure 3 legends are missing the information related to sample size and time point. Line 235 states that only top 25 genes were selected for graphical representation. Please mention the criterion of selection and also in my opinion presenting this information in table format is more advisable rather than selecting top 25.   Can authors improve figure 5 like figure 6? This figure hardly makes any sense rather than listing the genes and pathways. Section 3.6 includes lots of unexplained abbreviates, please correct them. Figure 7 legend mentions that independent samples T test was used to compare 6 independent samples/treatment groups. It’s quite confusing and not clear how it is possible? Please use the correct statistics to compare the groups and re-analyze the results. Line 286 reads, species is an important pathway for understanding thire speciation. Confusing, please correct?

In general the statistical analysis and parameters used is quite confusing. Thus it’s very hard to comprehend anything based on this analysis. I personally believe that after justifications to the answers and with improved analysis this manuscript will fare better chance to get published.  

Good Luck!!!

Author Response

Dear reviewer:

      I am very appreciating your review, and answer your questions as fellow:

Reviewer2: Article title doesn’t read well. Try to improve if possible like changing “Transcriptomics” I to “Transcriptomic analysis”.

Answer: we adjusted the article title as “Transcriptomic analysis reveals insect hormone biosynthesis pathway involved in desynchronized development phenomenon in hybridized sibling species of tea geometrids (Ectropis grisescens and Ectropis obliqua)”

Reviewer2: Lines 52, not clear what COI stands for? Make sure to use full forms of any abbreviations before denoting this”

Answer: we add the full form “Cytochrome Oxidase I (COI)” in first appeared in the manuscript.

Reviewer2: In my opinion, the final lines of introduction (line 76-77) belong to conclusion section.

Answer: we deleted the section which related to conclusion and changed another expression as “During the study, transcriptomic analysis was applied for those asynchronously growing hybrid individuals in order to explore the regulating mechanism of desynchronized development phenomenon.”

Reviewer2: Modify the line 85, Cytochrome Oxidase I (COI) needs to be mentioned before rather than here.

Answer: we made a modification as requirement of reviewer.

Reviewer2: I presume this article needs more information on statistical tests done and threshold used. Like, information of line 109-110, Normal data distribution was tested using Kolmogorov Smirnov test in SPSS should be moved to formal statistical analysis section.

Answer: we add a subtitle of “2.9. Statistical analysis” in part of Materials and Methods. Then, we picked the related contents into the section.

Reviewer2: The line 112 states that filial generation of E. grisescens ♀ × obliqua ♂ was used in this transcriptomic analysis. Any logical reason to select and use this cross and filial generation rather than E. grisescens ♀ × E. grisescens ♂?

Answer: we choose the filial generation of E. grisescens ♀ × obliqua ♂as the materials of transcriptomic analysis according to the result of hybridization experiment in 3.2 section. In the experiment, the filial generation of E. grisescens♀ × obliqua♂ showed a desynchronized development phenomenon. In comparison, the generation of E. grisescens♀ × E. grisescens♂ showed a normal development. So, we choose developing slow and fast samples as researching materials in the filial generation of E. grisescens ♀ × obliqua♂.

Reviewer2: As mentioned in line 148, authors chosen absolute value of log 2 (fold change) > 1 as threshold for DEG’s. Any specific reasons? As per general understanding the people use >2 fold change as threshold for this kind of analysis.

Answer: During the analysis of DEG’s, we found some used the log 2 (fold change) > 2 and some used the log 2 (fold change) >1 as the threshold such as (Aftab et al. 2019) and (liu, et al. 2019). We also tried to use the log 2 (fold change) > 2 as the threshold and found the DEG’s were less. We worried that threshold would submerge the key genes. So, we choose the the log 2 (fold change) >1 as the threshold.

Reference:

Jarwar, AR; Hao, K; Bitume, EV; Ullah, H; Cui, DG; Nong, XQ; Wang, GJ; Tu, XL; Zhang, ZH. Comparative transcriptomic analysis reveals molecular profiles of central nervous system in maternal diapause induction of Locusta migratoria. G3 Genesgenetics 2019, (9): 3287–3296.

Liu, PF; Wang, WW; Ling, XF; Lu, Q; Zhang, JW; He, R; Chen, H. Regulation of hormone-related genes in Ericerus pela (Hemiptera: Coccidae) for Dimorphic Metamorphosis. Journal of Insect Science 2019, 19(5): 16, 1–9.

Reviewer2: Again, figure 2 doesn’t explain enough the king of error bars used in graphing. I hope to include more information related graphing in figure legends.

Answer: we added the description of error bars in figure legends.

Reviewer2: Also in my opinion, sequence of figure needs to be changed to make it more presentable. Like, C & D should be A & B while A & B should be C & D.

Answer: we made a modification as requirement of reviewer.

Reviewer2: Figure 3 legends are missing the information related to sample size and time point.

Answer: we added the information related to sample size in Materials and Methods 2.4 and time point in Figure legends.

Reviewer2: Line 235 states that only top 25 genes were selected for graphical representation. Please mention the criterion of selection and also in my opinion presenting this information in table format is more advisable rather than selecting top 25.

Answer: we selected the top 25 genes refer to (Wang et al., 2019). Nevertheless, after changed the bar plot to bubble plot, we illustrated all significantly different GO terms in the new figure and picked some GO terms considerably high adjusted p-value listed. And the table of all genes and pathways were added into the supplementary files.

Reference: Wang, GB; Na, S; Duan, XX; Leng, ZM; Jiang, YR; Shi, SL; Yang, RS; Qi, L. Transcriptome sequencing to unravel the molecular mechanisms underlying the cuticle liquefaction of Antheraea pernyi following Antheraea pernyi nucleopolyhedrovirus challenge. Molecular Immunology 2019, 109, 108-115.

Reviewer2: Can authors improve figure 5 like figure 6? This figure hardly makes any sense rather than listing the genes and pathways.

Answer: we made a modification as requirement of reviewer.

Reviewer2: Section 3.6 includes lots of unexplained abbreviates, please correct them.

Answer: we made a modification as requirement of reviewer.

Reviewer2: Figure 7 legend mentions that independent samples T test was used to compare 6 independent samples/treatment groups. It’s quite confusing and not clear how it is possible? Please use the correct statistics to compare the groups and re-analyze the results.

Answer: the significantly different expression genes were re-analyzed by the standardized pipeline of DESeq2 in R, which results were added into the supplementary files.

Reviewer2: Line 286 reads, species is an important pathway for understanding thire speciation. Confusing, please correct?

Answer: we made a modification and changed another expression as “geographical distribution is an important aspect for deducing the sibling species formation and differentiation”.

Reviewer 3 Report

Dear Authors,

The manuscript of Zhibo Wang et al. presents a comprehensive transcriptomic analysis of the genes expression involved in the moulting hormone biosynthetic pathway controlling the insects hybrids showing desynchronized development This type of transcriptomic analysis can be further applied to explore other   mechanisms of insect development using other species.

I recommend the manuscript to be considered for publication in the 'Insects" journal without the need for other experiments.

However, I would recommend to have the ML tree of tea geometrids and the outgroup species based on COI sequences-from Supplementary fig 1 to be included in the main body of the manuscript as a main figure.

Author Response

Dear reviewer:

      I am very appreciating your review, and answer your questions as fellow:

Reviewer3: I would recommend to have the ML tree of tea geometrids and the outgroup species based on COI sequences-from Supplementary fig 1 to be included in the main body of the manuscript as a main figure.

Answer: we made a modification according to the advice of reviewer, putting the supplementary fig 1 to the main body of the manuscript as fig 1.

Round 2

Reviewer 1 Report

I have no further remarks.

Reviewer 2 Report

Article got improved remarkably after revision and thus I recommend that the paper can be accepted for the publication depending upon the discretion of other reviewers and editors.

Good Luck!!!